# *GOAt*: Explaining Graph Neural Networks via Graph Output Attribution

## Abstract

Understanding the decision-making process of Graph Neural Networks (GNNs) is crucial to their interpretability. Present methods for explaining GNNs typically rely on training auxiliary models, and may struggle with issues such as overfitting to noise, insufficient discriminability, and inconsistent explanations across data samples of the same class. This paper introduces Graph Output Attribution (GOAt), a novel method to attribute graph outputs to input graph features, creating GNN explanations that are faithful, discriminative, as well as stable across similar samples. By expanding the GNN as a sum of scalar products involving node features, edge features and activation patterns, we propose an efficient analytical method to compute contribution of each node or edge feature to each scalar product and aggregate the contributions from all scalar products in the expansion form to derive the importance of each node and edge. Through extensive experiments on synthetic and real data, we show that our method has consistently outperformed various state-of-the-art GNN explainers in terms of fidelity, discriminability, and stability.

## 1 Introduction

Graph Neural Networks (GNNs) have demonstrated notable success in learning representations from graph-structured data in various fields [14, 9, 30]. However, their black-box nature has driven the need for explainability, especially in sectors where transparency and accountability are essential, such as finance [28], healthcare [1], and security [18]. The ability to interpret GNNs can provide insights into the mechanisms underlying deep models and help establish trustworthy predictions.

Existing attempts to explain GNNs usually focus on local-level or global-level explainability. Local-level explainers [31, 17, 21, 27, 10, 15, 23, 3] typically train a secondary model to identify the critical graph structures that best explain the behavior of a pretrained GNN for specific input instances. These methods are always optimized for ground-truth explanations or fidelity metrics, yet may not be able to generate consistent explanations for similar graph samples or produce accurate and human-intelligible explanations for class discrimination. Global-level explainers [2, 11] perform prototype learning or random walk on the explanation instances to extract the global explanations over a multitude of graph samples. However, their effectiveness rely heavily on the quality of local-level explanations.

In this paper, we introduce a computationally efficient local-level GNN explanation technique called **G**raph **O**utput **At**tribution (*GOAt*) to overcome the limitations of existing methods. Unlike methods that rely on back-propagation with gradients [20, 4, 22, 8] and those relyinig on hyper-parameters or training complex black-box models [17, 15, 23, 3], our approach enables attribution of GNN output to input features, leveraging the repetitive sum-product structure in the forward pass of a GNN.

Given that the matrix multiplication in each GNN layer adheres to linearity properties and the activation functions operate element-wise, a GNN can be represented in an expansion form as a

sum of scalar product terms, involving input graph features, model parameters, as well as *activation patterns* that indicate the activation levels of the scalar products. Based on the notion that all scalar variables $X_i$ in a scalar product term $g = cX_1X_2 \ldots X_N$ contribute equally to $g$, where $c$ is a constant, we can attribute each product term to its corresponding factors and thus to input features, obtaining the importance of each node or edge feature in the input graph to GNN outputs. We present case studies that demonstrate the effectiveness of our analytical explanation method *GOAt* on typical GNN variants, including GCN, GraphSAGE, and GIN.

Besides the fidelity metric, which is commonly used to assess the faithfulness of GNN explanations, we introduce two new metrics to evaluate the *discriminability* and *stability* of the explanation, which are under-investigated by prior literature. Discriminability refers to the ability of explanations to distinguish between classes, which is assessed by the difference between the mean explanation embeddings of different classes, while stability refers to the ability to generate consistent explanations across similar data instances, which is measured by the percentage of data samples covered by top-$k$ explanations. Through comprehensive experiments based on on both synthetic and real-world datasets along with qualitative analysis, we show the outstanding performance of our proposed method, *GOAt*, in providing highly faithful, discriminative, and stable explanations for GNNs, as compared to a range of state-of-the-art methods.

## 2 Problem Formulation

**Graph Neural Networks** Let $G = (\mathcal{V}, \mathcal{E})$ be a graph, where $\mathcal{V} = \{v_1, v_2, \ldots, v_N\}$ denotes the set of nodes and $\mathcal{E} \subseteq \mathcal{V} \times \mathcal{V}$ denotes the set of edges. The node feature matrix of the graph is represented by $X \in \mathbb{R}^{N \times d}$, and the adjacency matrix is represented by $A \in \{0,1\}^{N \times N}$ such that $A_{ij} = 1$ if there exists an edge between nodes $v_i$ and $v_j$. The task of a GNN is to learn a function $f(G)$, which maps the input graph $G$ to a target output, such as node labels, graph labels, or edge labels. Formally speaking, for a given GNN, the hidden state $h_i^{(l)}$ of node $v_i$ at its layer $l$ can be represented as:

$$h_i^{(l)} = \text{COMBINE}^{(l)} \left\{ h_i^{(l-1)}, \text{AGGREGATE}^{(l)} \left( \left\{ h_j^{(l-1)}, \forall v_j \in \mathcal{N}_i \right\} \right) \right\}, \tag{1}$$

where $\mathcal{N}_i$ represents the set of neighbors of node $v_i$ in the graph. $\text{COMEBINE}^{(l)}(\cdot)$ is a COMBINE function such as concatenation [9], while $\text{AGGREGATE}^{(l)}(\cdot)$ are AGGREGATE functions with aggregators such as ADD. We focus on GNNs that adopt the non-linear activation function ReLU in COMBINE or AGGREGATE functions.

**Local-level GNN Explainability** Our goal is to generate a faithful explanation for a graph instance $G = (\mathcal{V}, \mathcal{E})$ by identifying a subset of edges $S \subseteq \mathcal{E}$, given a GNN $f(\cdot)$ pretrained on a set of graphs $\mathcal{G}$. The term *faithful* refers to the explanation's ability to perform well in not only *fidelity* [33] and *robustness* [3] metrics, but also *stability* in identifying consistent patterns. We highlight edges instead of nodes as suggested by [7] that edges have more fine-grained information than nodes while giving human-understandable explanations like subgraphs.

## 3 Method

This section begins by presenting our fundamental definition of equal contribution in a product term and its application in an example of a toy graph neural network. Then, we mathematically present *GOAt* method for explaining typical GNNs, followed by a case study on GCN [14]. Additional case studies of applying *GOAt* to GraphSAGE [9] and GIN [30] are included in the Appendix.

### 3.1 Definitions

Consider a function $g(X_1, \ldots, X_M)$ of $M$ variables $\mathbf{X} = \{X_1, \ldots, X_M\}$. If we let a pair of variables $(X_i, X_j)$ be set to $(X_i, X_j) = (x_i, x_j)$, we will obtain a manifold $g_{X_i=x_i, X_j=x_j}(\mathbf{X} \backslash \{X_i, X_j\})$, which represents $g(\cdot)$ when all variables excluding $X_i$ and $X_j$ can vary. Consider a base manifold $g_{X_i=x_i', X_j=x_j'}(\mathbf{X} \backslash \{X_i, X_j\})$. If we can obtain two identical manifolds by setting $(X_i, X_j) = (x_i, x_j')$ and $(X_i, X_j) = (x_i', x_j)$, it will indicate that changing $X_i = x_i'$ to $X_i = x_i$ is equivalent to changing $X_j = x_j'$ to $X_j = x_j$ with respect to the base manifold at $(X_i, X_j) = (x_i', x_j')$. For example, a function $g(x, y, z) = 2xy + x^2z$ has three variables $x, y, z$, we consider taking

83 $g_{x=-1,y=-1}(z) = z + 2$ as the base manifold. Since $g_{x=1,y=-1}(z) = g_{x=-1,y=1}(z) = z - 2$, we
84 say that changing $x = -1$ to $x = 1$ is equivalent to changing $y = -1$ to $y = 1$ with respect to the
85 base manifold at $(x, y) = (-1, -1)$.

86 **Definition 1** (**Equal Contribution**). *Given a function $g(\mathbf{X})$ where $\mathbf{X} = \{X_1, \ldots, X_M\}$ represents*
87 *$M$ variables, we say that variables $X_i$ and $X_j$ have equal contribution to function $g$ at $(x_i, x_j)$*
88 *with respect to the base manifold at $(x_i', x_j')$ if and only if setting $X_i = x_i, X_j = x_j'$ and setting*
89 *$X_i = x_i', X_j = x_j$ yield the same manifold, i.e.,*

$$g_{X_i=x_i,X_j=x_j'}(\mathbf{X}\backslash\{X_i, X_j\}) = g_{X_i=x_i',X_j=x_j}(\mathbf{X}\backslash\{X_i, X_j\})$$

90 *for any values of $\mathbf{X}$ excluding $X_i$ and $X_j$.*

91 **Lemma 2** (**Equal Contribution in a product**). *Given a function $g(\mathbf{X})$ defined as $g(\mathbf{X}) =$*
92 *$b\prod_{k=1}^{M} X_k$, where $b$ is a constant, and $\mathbf{X} = \{X_1, \ldots, X_M\}$ represents $M$ uncorrelated variables.*
93 *Each variable $X_k$ is either $0$ or $x_k$, depending on the absence or presence of a certain feature. Then,*
94 *all the variables in $\mathbf{X}$ contribute equally to $g(\mathbf{X})$ at $[x_1, \ldots, x_M]$ with respect to $[0, \ldots, 0]$.*

95 Proofs of all Lemmas and Theorems can be found in the Appendix. Since all the binary variables
96 have equal contribution, we define the contribution of each variable $X_k$ to $g(\mathbf{X}) = b\prod_{k=1}^{M} X_k$ for
97 all $k = 1, \ldots, M$, as

$$I_{X_k} = \frac{b\prod_{i=1}^{M} x_i}{M}. \tag{2}$$

98 For example, let $f(A, X) = AXW$ be a simple 2-node GNN for node classification, where $A, X, W$
99 are $2 \times 2$ matrices that denote adjacency matrix, node feature matrix, and weight matrix, respectively.
100 Then, we can represent each entry in the resulting $2 \times 2$ matrix $f(A, X)$ as an expansion form:

$$f_{i,j}(A, X) = \sum_{k=0}^{1} \sum_{l=0}^{1} A_{i,k} X_{k,l} W_{l,j}, \tag{3}$$

101 where $f_{i,j}(A, X)$ represents the prediction of the $i$-th node for the $j$-th class. In a pretrained
102 GNN, parameter $W$ is fixed. Thus, only $A_{i,k}$ and $X_{k,l}$ contribute to the value of each *scalar product*
103 $A_{i,k} X_{k,l} W_{l,j}$. As $A_{i,k}$ is usually independent of $X_{k,l}$ under proper data cleaning, we can calculate the
104 contributions of $A_{i,k}$ and $X_{k,l}$ to the scalar product $A_{i,k} X_{k,l} W_{l,j}$ by $I_{A_{i,k}} = I_{X_{k,l}} = \frac{1}{2} A_{i,k} X_{k,l} W_{l,j}$
105 based on Lemma 2 and Equation (2). By similar computations for all the scalar products in the
106 expansion form of $f(\cdot)$, we can obtain the contribution of all the input features to each entry of the
107 output matrix.

### 3.2 Explaining Graph Neural Networks via Attribution

109 A typical GNN [14, 9, 30] for node or graph classification tasks usually comprises 2-6 message-
110 passing layers for learning node or graph representations, followed by several fully connected layers
111 that serve as the classifier. With the hidden state $h_i^{(l)}$ of node $v_i$ at the $l$-th message-passing layer
112 defined as Equation (1), we generally have the hidden state $H^{(l)}$ of a data sample as:

$$H^{(l)} = \sigma\left(\Phi^{(l)}\left(\left(A + \epsilon^{(l)} I\right) H^{(l-1)}\right) + \lambda \Psi^{(l)}\left(H^{(l-1)}\right)\right), \tag{4}$$

113 where $A$ is the adjacency matrix, $\epsilon^{(l)}$ refers to the self-loop added to the graph if fixed to 1, otherwise it
114 is a learnable parameter, $\sigma(\cdot)$ is the element wise activation function, $\Phi^{(l)}$ and $\Psi^{(l)}$ can be Multilayer
115 Perceptrons (MLP) or linear mappings, $\lambda \in \{0, 1\}$ determines whether a concatenation is required.
116 If the COMBINE step of a GNN requires a concatenation, we have $\lambda = 1$ and $\epsilon^{(l)} = 1$; if the
117 COMBINE step requires a weighted sum, we have $\epsilon^{(l)}$ set trainable and $\lambda = 0$. Alternatively,
118 Equation (4) can be expanded to:

$$H^{(l)} = \sigma\left(AH^{(l-1)}\prod_{k=1}^{K} W^{\Phi_k^{(l)}} + \epsilon^{(l)} H^{(l-1)}\prod_{k=1}^{K} W^{\Phi_k^{(l)}} + \lambda H^{(l-1)}\prod_{q=1}^{Q} W^{\Psi_q^{(l)}}\right), \tag{5}$$

119 where $K, Q$ refer to the number of MLP layers in $\Phi^{(l)}(\cdot)$ and $\Psi^{(l)}(\cdot)$, and $W^{\Phi_k^{(l)}}$ and $W^{\Psi_q^{(l)}}$ are the
120 trainable parameters in $\Phi_k^{(l)}$ and $\Psi_q^{(l)}$.

121 Given a certain data sample and a pretrained GNN, for an element-wise activation function we can
122 define the activation pattern as the ratio between the output and input of the activation function:

**Definition 3 (Activation Pattern).** *Denote $H^{(l)'}$ and $H^{(l)}$ as the hidden representations before and after passing through the element-wise activation function at the $l$-th layer, we define activation pattern $P^{(l)}$ for a given data sample as*

$$P_{i,j}^{(l)} = \begin{cases} \dfrac{H_{i,j}^{(l)}}{H_{i,j}^{(l)'}}, & \text{if } H_{i,j}^{(l)'} \neq 0 \\ 0, & \text{otherwise} \end{cases}$$

*where $P_{i,j}^{(l)}$ is the element-wise activation pattern for the $j$-th feature of $i$-th node at layer $l$.*

Hence, the hidden state $H^{(l)}$ at the $l$-th layer for a given sample can be written as

$$H^{(l)} = P^{(l)} \odot \left( AH^{(l-1)} \prod_{k=1}^{K} W^{\Phi_k^{(l)}} + \epsilon^{(l)} H^{(l-1)} \prod_{k=1}^{K} W^{\Phi_k^{(l)}} + \lambda H^{(l-1)} \prod_{q=1}^{Q} W^{\Psi_q^{(l)}} \right), \quad (6)$$

where $\odot$ represents element-wise multiplication. Thus, similar to Equation (3), we can expand the expression of each output entry in a GNN $f(A, X)$ into a sum of scalar products, where each scalar product is the multiplication of corresponding entries in $A$, $X$, $W$, and $P$ in all layers. Then each scalar product can be written as

$$z = \mathbb{C} \cdot \left( P_{\alpha_{10},\beta_{11}}^{(1)} \dots P_{\alpha_{L0},\beta_{L1}}^{(L)} \right) \left( P_{\alpha_{L0},\gamma_{11}}^{(c_1)} \dots P_{\alpha_{L0},\gamma_{(M-1)1}}^{(c_{(M-1)})} \right) \cdot \\ \left( A_{\alpha_{L0},\alpha_{L1}}^{(L)} \dots A_{\alpha_{10},\alpha_{11}}^{(1)} \right) X_{i,j} \left( W_{\beta_{10},\beta_{11}}^{(1)} \dots W_{\beta_{L0},\beta_{L1}}^{(L)} \right) \left( W_{\gamma_{10},\gamma_{11}}^{(c_1)} \dots W_{\gamma_{M0},\gamma_{M1}}^{(c_M)} \right), \quad (7)$$

where $\mathbb{C}$ is a constant, $c_k$ refers to the $k$-th layer of the classifier, $(\alpha_{l0}, \alpha_{l1}), (\beta_{l0}, \beta_{l1}), (\gamma_{l0}, \gamma_{l1})$ are *(row, column)* indices of the corresponding matrices at layer $l$. In a classifier with $M$ MLP layers, only $(M-1)$ layers adopt activation functions. Therefore, we do not have $P_{\alpha_{L0},\gamma_{M1}}^{(c_M)}$ in Equation (7). For scalar products without factors of $A$, all $A$'s are considered as constants equal to 1 in Equation (7). Since the GNN model parameters are pretrained and fixed, we only consider $A$, $X$, and all the $P$ terms as the variables in each product term.

**Lemma 4 (Equal Contribution variables in the GNN expansion form's scalar product).** *For a scalar product term $z$ in the expansion form of a pretrained GNN $f(\cdot)$, when the number of nodes $N$ is large, all variables in $z$ have equal contributions to the scalar product $z$.*

Hence, by Equation (2), we can calculate the contribution $I_\nu(z)$ of a variable $\nu$ (i.e., an entry in $A$, $X$ and $P$ matrices) to each scalar product $z$ (given by Equation (7)) by:

$$I_\nu(z) = \frac{z}{|V(z)|}, \quad (8)$$

where function $V(\cdot)$ represents the set of variables in its input, and $|V(z)|$ denotes the number of unique **variables** in $z$, e.g., $V(x^2 y) = \{x, y\}$, and $|V(x^2 y)| = 2$.

Similar to Section 3.1, an entry $f_{m,n}(A, X)$ of the output matrix $f(A, X)$ can be expressed by the sum of all the related scalar products as

$$f_{m,n}(A, X) = \sum \mathbb{C} \cdot \left( P_{\alpha_{10},\beta_{11}}^{(1)} \dots P_{m,\beta_{L1}}^{(L)} \right) \cdot \left( P_{m,\gamma_{11}}^{(c_1)} \dots P_{m,\gamma_{(M-1)1}}^{(c_{(M-1)})} \right) \cdot \left( A_{m,\alpha_{L1}}^{(L)} \dots A_{\alpha_{10},\alpha_{11}}^{(1)} \right) \\ \cdot X_{i,j} \cdot \left( W_{\beta_{10},\beta_{11}}^{(1)} \dots W_{\beta_{L0},\beta_{L1}}^{(L)} \right) \cdot \left( W_{\gamma_{10},\gamma_{11}}^{(c_1)} \dots W_{\gamma_{M0},n}^{(c_M)} \right), \quad (9)$$

where summation is over all possible $(\alpha_{l0}, \alpha_{l1}), (\beta_{l0}, \beta_{l1}), (\gamma_{l0}, \gamma_{l1})$, for message passing layer $l = 1, \dots, L$ or classifier layer $l = 1, \dots, M$, as well as all $i, j$ indices for $X$. By summing up the contribution of each variable $\nu$ among the entries in the $A$, $X$ and $P$'s in all the scalar products in the expansion form of $f_{m,n}(\cdot)$, we can obtain the contribution of $\nu$ to $f_{m,n}(\cdot)$ as:

$$I_\nu(f_{m,n}(\cdot)) = \sum_{z \text{ in } f_{m,n}(\cdot) \text{ that contain } \nu} \frac{z}{|V(z)|}. \quad (10)$$

**Theorem 5 (Contribution of variables in the expansion form of a pretrained GNN).** *Given Equations (8) and (10), for each variable $\nu$ (i.e., an entry in $A$, $X$ and $P$ matrices), when the number of nodes $N$ is large, we can approximate $I_\nu(f_{m,n}(\cdot))$ by:*

$$I_\nu(f_{m,n}(\cdot)) = \sum_{z \text{ in } f_{m,n}(\cdot) \text{ that contain } \nu} \frac{O(\nu, z)}{\sum_{\rho \text{ in } z} O(\rho, z)} \cdot z, \quad (11)$$

*where $O(\nu, z)$ denotes the number of occurrences of $\nu$ among the variables of $z$.*

Recall that $|V(z)|$ stand for the number of **unique variables** in $z$. Hence the total number of occurrences of all the variables $\sum_{\rho \text{ in } z} O(\rho, z)$ is not necessarily equal to $|V(z)|$. For example, if all of $\{A^{(1)}_{\alpha_{10},\alpha_{11}}, \ldots, A^{(L)}_{\alpha_{L0},\alpha_{L1}}\}$ in $z$ are unique entries in $A$, then they can be considered as $L$ independent variables in the function representing $z$. If two of these occurrences of variables refer to the same entry in $A$, then there are only $(L-1)$ unique variables related to $A$.

Although Theorem 5 gives the contribution of each entry in $A$, $X$ and $P$'s, we need to further attribute $P$'s to $A$ and $X$ and allocate the contribution of each activation pattern $P^{(r)}_{a,b}$ to node features $X$ and edges $A$ by considering all non-zero features in $X_a$ of node $v_a$ and the edges within $m$ hops of node $v_a$, as these inputs may contribute to the activation pattern $P^{(r)}_{a,b}$. However, determining the exact contribution of each feature that contributes to $P^{(r)}_{a,b}$ is not straightforward due to non-linear activation. We approximately attribute all relevant features equally to $P^{(r)}_{a,b}$. That is, each input feature $\nu$ that has nonzero contribution to $P^{(r)}_{a,b}$ will share an equal contribution of $I_{P^{(r)}_{a,b}}(f_{m,n}(\cdot))/|V(P^{(r)}_{a,b})|$, where $|V(P^{(r)}_{a,b})|$ denotes the number of distinct node and edge features in $X$ and $A$ contributing to $P^{(r)}_{a,b}$, which is exactly all non-zero features in $X_a$ of node $v_a$ and the adjacency matrix entries within $r$ hops of node $v_a$. Finally, based on Equation (11), we can obtain the contribution of an input feature $\nu$ in $X$, $A$ of a graph instance to the $(m,n)$-th entry of the GNN output $f(\cdot)$ as:

$$\widehat{I}_\nu(f_{m,n}(\cdot)) = I_\nu(f_{m,n}(\cdot)) + \sum_{P^{(r)}_{a,b} \text{ in } f_{m,n}(\cdot), \text{ with } \nu \text{ in } P^{(r)}_{a,b}} \frac{I_{P^{(r)}_{a,b}}(f_{m,n}(\cdot))}{|V(P^{(r)}_{a,b})|}, \tag{12}$$

where $\nu$ is an entry in the adjacency matrix $A$ or the input feature matrix $X$, $P^{(r)}_{a,b}$ denotes an entry in all the activation patterns. Thus, we have attributed $f(\cdot)$ to each input feature of a given data instance.

Our approach meets the *completeness* axiom, which is a critical requirement in attribution methods [25, 24, 6]. This axiom guarantees that the attribution scores for input features add up to the difference in the GNN's output with and without those features. Passing this sanity check implies that our approach provides a more comprehensive account of feature importance than existing methods that only rank the top features [3, 17, 20, 31, 27, 23].

## 3.3 Case Study: Explaining Graph Convolutional Network (GCN)

GCNs use a simple sum in the combination step, and the adjacency matrix is normalized with the diagonal node degree matrix $D$. Hence, the hidden state of a GCN's $l$-th message-passing layer is:

$$H^{(l)} = \text{ReLU}\left(V H^{(l-1)} W^{(l)} + B^{(l)}\right), \tag{13}$$

where $V = D^{-\frac{1}{2}}(A+I)D^{-\frac{1}{2}}$ represents the normalized adjacency matrix with self-loops added. Suppose a GCN has three convolution layers and a 2-layer MLP as the classifier, then its expansion form without the activation functions $\text{ReLU}(\cdot)$ will be:

$$
\begin{aligned}
f(V,X)_{\mathbf{P}} = {}& V^{(3)}V^{(2)}V^{(1)}XW^{(1)}W^{(2)}W^{(3)}W^{(c_1)}W^{(c_2)} + V^{(3)}V^{(2)}B^{(1)}W^{(2)}W^{(3)}W^{(c_1)}W^{(c_2)} \\
& + V^{(3)}B^{(2)}W^{(3)}W^{(c_1)}W^{(c_2)} + B^{(3)}W^{(c_1)}W^{(c_2)} + B^{(c_1)}W^{(c_2)} + B^{(c_2)},
\end{aligned} \tag{14}
$$

where $V^{(l)} = V$ is the normalized adjacency matrix in the $l$-th layer's calculation. In the actual expansion form with the activation patterns, the corresponding $P^{(m)}$'s are multiplied whenever there is a $W^{(m)}$ or $B^{(m)}$ in a scalar product, excluding the last layer $W^{(c_2)}$ and $B^{(c_2)}$. For example, in the scalar products corresponding to $V^{(3)}V^{(2)}V^{(1)}XW^{(1)}W^{(2)}W^{(3)}W^{(c_1)}W^{(c_2)}$, there are eight variables consisting of four $P$'s, one $X$, and three $V$'s. By Equation (11), an adjacency entry $V_{i,j}$ itself will contribute $\frac{1}{8}$ of $p(V^{(3)}V^{(2)}_{:i}V^{(1)}_{i,j}X_{j:}W^{(1)}W^{(2)}W^{(3)}W^{(c_1)}W^{(c_2)}) + p(V^{(3)}_{:i}V^{(2)}_{i,j}V^{(1)}_{j:}XW^{(1)}W^{(2)}W^{(3)}W^{(c_1)}W^{(c_2)}) + p(V^{(3)}_{i,j}V^{(2)}_{j:}V^{(1)}XW^{(1)}W^{(2)}W^{(3)}W^{(c_1)}W^{(c_2)})$, where $p(\cdot)$ denotes the results with the element-wise multiplication of the corresponding activation patterns applied at the appropriate layers. After we obtain the contribution of $V_{i,j}$ itself on all the scalar products, we can follow Equation (12) to allocate the contribution of activation patterns to $V_{i,j}$.

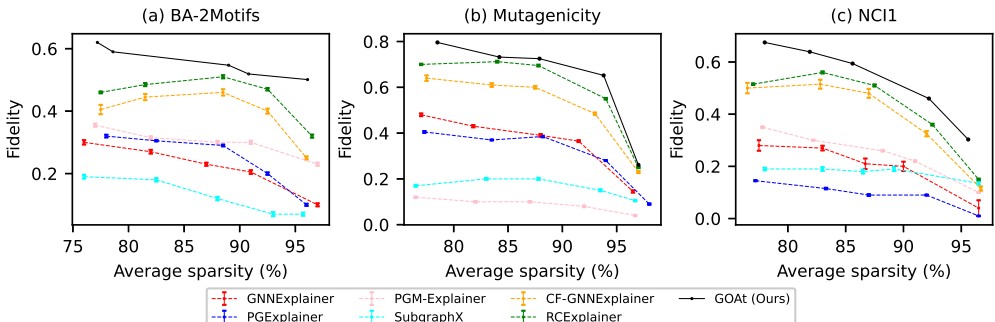

Figure 1: Fidelity performance averaged across 10 runs on the pretrained GCNs for the datasets at different levels of average sparsity.

With Equation (14), we find that when both $V$ and $X$ are set to zeros, $f(\cdot)$ remains non-zero and is:

$$f(\mathbf{0}, \mathbf{0}) = p(B^{(3)}W^{(c_1)}W^{(c_2)}) + p(B^{(c_1)}W^{(c_2)}) + B^{(c_2)}, \tag{15}$$

where $B^{(c_2)}$ is the global bias, and the other terms have non-zero entries at the activated neurons. In other words, certain GNN neurons in the 3-rd and $c_1$-th layers may already be activated prior to any input feature being passed to the network. When we do feed input features, some of these neurons may remain activated or be toggled off. With Equation (12), we consider taking all 0's of the $X$ entries, $V$ entries and $P$ entries as the base manifold. Now, given that some of the $P$ entries in GCN are non-zero when all $X$ and $V$ set to zeros, as present in Equation (15), we will need to subtract the contribution of each features on these $P$ from the contribution values calculated by Equation (12). We let $\mathbf{P}'$ represent the activation patterns of $f(\mathbf{0}, \mathbf{0})$, then the calibrated contribution $\widehat{I}_{V_{i,j}}^{\text{cali}}(f(\cdot))$ of $V_{i,j}$ is given by:

$$\widehat{I}_{V_{i,j}}^{\text{cali}}(f(\cdot)) = \widehat{I}_{V_{i,j}}(f(V, X)) - \sum_{P_{a,b}'^{(r)} \text{ in } f(\mathbf{0},\mathbf{0}), \text{ with } V_{i,j} \text{ in } P_{a,b}'^{(r)}} \frac{I_{P_{a,b}'^{(r)}}(f(\mathbf{0}, \mathbf{0}))}{|V(P_{a,b}^{(r)})|}. \tag{16}$$

In graph classification tasks, a pooling layer such as mean-pooling is added after the convolution layers to obtain the graph representation. To determine the contribution of each input feature, we can simply apply the same pooling operation as used in the pre-trained GCN.

As we mentioned in Section 2, we aim to obtain the explanations by the critical edges in this paper, since edges have more fine-grained information than nodes. Therefore, we treat the edges as variables, while considering the node features $X$ as constants similar to parameters $W$ or $B$. This setup naturally aggregates the contribution of node features onto edges. By leveraging edge attributions, we are able to effectively highlight motifs within the graph structure.

## 4 Experiments

We conduct a series of experiments on the fidelity, discriminability and stability metrics to compare our method with the state-of-the-art methods including GNNExplainer [31], PGExplainer [17], PGM-Explainer [27], SubgraphX [33], CF-GNNExplainer [16], RCExplainer [3], RG-Explainer [23] and DEGREE [8]. As outlined in Section 2, we highlight edges as explanations as suggested by [7]. For baselines that identify nodes or subgraphs as explanations, we adopt the evaluation setup from [3].

We evaluate the performance of explanations on three variants of GNNs, which are GCN [14], GraphSAGE [9] and GIN [30]. The experiments are conducted on both the graph classification task and the node classification task. For graph classification task, we evaluate on a synthetic dataset, BA-2motifs [17], and two real-world datasets, Mutagenicity [13] and NCI1 [19]. For node classification task, we evaluate on three synthetic datasets [17], which are BA-shapes, BA-Community and Tree-grid. As space is limited, we will only present the key results here. Fidelity results on GIN and GraphSAGE, as well as the results of node classification tasks can be found in the Appendix. Discussions on the controversial metrics such as accuracy are also moved to the Appendix.

### 4.1 Fidelity

Fidelity [20, 32, 29, 3] is the decrease of predicted probability between original and new predictions after removing important edges, which are used to evaluate the faithfulness of explanations. It is

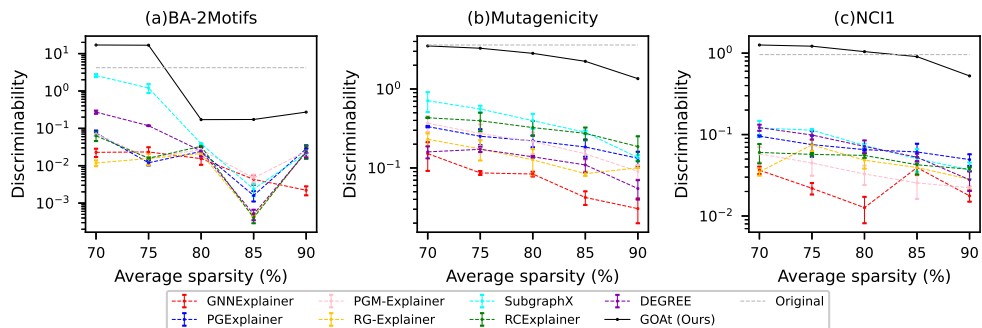

Figure 2: Discriminability performance averaged across 10 runs of the explanations produced by various GNN explainers at different levels of sparsity. "Original" refer to the performance of feeding the original data into the GNN without any modifications or explanations applied.

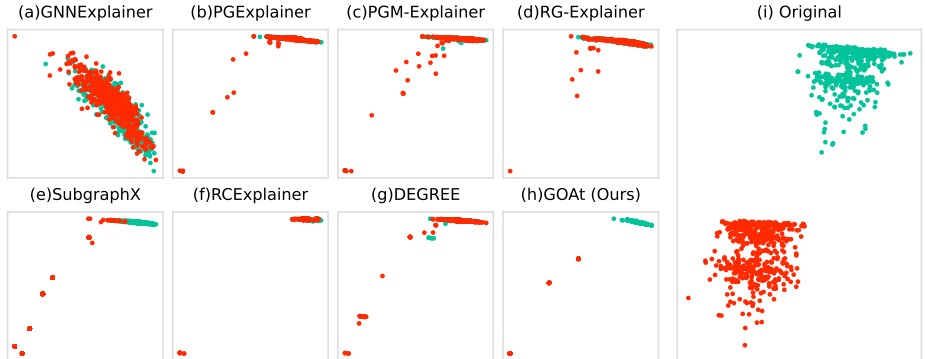

Figure 3: Visualization of explanation embeddings on the BA-2Motifs dataset. Subfigure (i) refers to the visualization of the original embeddings by directly feeding the original data into the GNN without any modifications or explanations applied.

defined as $fidelity(S, G) = f_y(G) - f_y(G \backslash S)$. As pointed out by [32], the fidelity may be sensitive to sparsity of explanations. The sparsity of an explanation $S \subseteq \mathcal{E}$ for a graph $G = \{\mathcal{V}, \mathcal{E}\}$ is given by $sparsity(S, G) = 1 - \frac{|S|}{|\mathcal{E}|}$. It indicates the percentage of edges that remain in $G$ after the removal of edges in $S$. Higher sparsity means fewer edges are identified as critical, which may have a smaller impact on the prediction probability. Therefore, we compare fidelity performance under similar levels of average sparsity, as in [33, 29, 3]. Figure 1 displays the fidelity results, with the baseline results sourced from [3]. Our proposed approach, *GOAt*, consistently outperforms the baselines in terms of fidelity across all sparsity levels, validating its superior performance in generating accurate and reliable faithful explanations. Among the other methods, RCExplainer exhibits the highest fidelity, as it is specifically designed for fidelity optimization. Notably, unlike the other methods that require training and hyperparameter tuning, *GOAt* offers the advantage of being a training-free approach, thereby avoiding any errors across different runs.

## 4.2 Discriminability

Discriminability, also known as discrimination ability [5, 12], refers to the ability of the explanations to distinguish between the classes. We define the discriminability between two classes $c_1$ and $c_2$ as the L2 norm of the difference between the mean values of explanation embeddings of the two classes. The embeddings used for explanations are taken prior to the last-layer classifier, with node embeddings employed for node classification tasks and graph embeddings utilized for graph classification tasks. In this procedure, only the explanation subgraph $S$ is fed into the GNN instead of $G$.

We show the discriminability across various sparsity levels on GCN, as illustrated in Figure 2. Due to the significant performance gap between the baselines and *GOAt*, a logarithmic scale is employed. Our approach consistently outperforms the baselines in terms of discriminability across all sparsity levels, demonstrating its superior ability to generate accurate and reliable class-specific explanations. Notably, at $sparsity = 0.7$, *GOAt* achieves higher discriminability than the original graphs on the BA-2Motifs and NCI1 datasets. This indicates that *GOAt* effectively reduces noise unrelated to the investigated class while producing informative class explanations. Additionally, we observe a

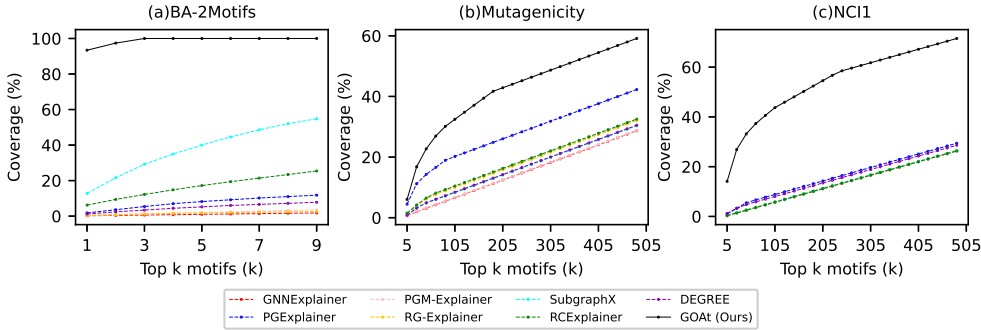

Figure 4: Coverage of the top-$k$ explanations across the datasets.

substantial decrease in discriminability between sparsity levels of 0.75 and 0.8 on BA-2Motifs. This implies that a minimum of approximately 25% of the edges is necessary to distinguish between the classes, which is in line with our expectation, given that a "house" motif, consisting of 6 edges, usually represents 24% of the total edges (on average, the total number of edges in BA-2Motifs is 25).

Furthermore, we present scatter plots to visualize the explanation embeddings generated by various GNN explainers. Figure 3 showcases the explanation embeddings obtained from different GNN explaining methods on the BA-2Motifs dataset, with $sparsity = 0.7$. More scatter plots on Mutagenicity and NCI1 and can be found in the Appendix. The explanations generated by GNNExplainer fail to exhibit class discrimination, as all the data points are clustered together without any distinct separation. While some of the Class 1 explanations produced by PGExplainer, PGM-Explainer, RG-Explainer, RCExplainer, and DEGREE are noticeably separate from the Class 0 explanations, the majority of the data points remain closely clustered together. As for SubgraphX, most of its Class 1 explanations are isolated from the Class 0 explanations, but there is a discernible overlap between the Class 1 and Class 0 data points. In contrast, our method, *GOAt*, generates explanations that clearly and effectively distinguish between Class 0 and Class 1, with no overlapping points and a substantial separation distance, highlighting the strong discriminability of our approach.

### 4.3 Stability of extracting motifs

As we will later show in Section 4.4, it is often observed that datasets contain specific class motifs. For instance, in the BA-2Motifs dataset, the Class 1 motif exhibits a "house" structure. To ensure the stability of GNN explainers in capturing the class motifs across diverse data samples, we aim for the explanation motifs to exhibit relative consistency for data samples with similar properties, rather than exhibiting significant variations. To quantify this characteristic, we introduce the *stability* metric, which measures the coverage of the top-$k$ explanations across the dataset. An ideal explainer should generate explanations that cover a larger number of data samples using fewer motifs. This characteristic is also highly desirable in global-level explainers, such as [2, 11]. We illustrate the stability of the unbiased class as the percentage converge of the top-$k$ explanations produced on GCN with $sparsity = 0.7$ in Figure 4. Our approach surpasses the baselines by a considerable margin in terms of the stability of producing explanations. Specifically, *GOAt* is capable of providing explanations for all the Class 1 data samples using only three explanations. This explains why there are only three Class 1 scatters visible in Figure 3.

### 4.4 Qualitative analysis

We present the qualitative results of our approach in Table 1, where we compare it with state-of-the-art baselines such as PGExplainer, SubgraphX, and RCExplainer. The pretrained GNN achieves a 100% accuracy on the BA-2Motifs dataset. As long as it successfully identifies one class, the remaining data samples naturally belong to the other class, leading to a perfect accuracy rate. Based on the explanations from *GOAt*, we have observed that the GNN effectively recognizes the "house" motif that is associated with Class 1. In contrast, other approaches face difficulties in consistently capturing this motif. The Class 0 motifs in the Mutagenicity dataset generated by *GOAt* represent multiple connected carbon rings. This indicates that the presence of more carbon rings in a molecule increases its likelihood of being mutagenic (Class 0), while the presence of more "C-H" or "O-H" bonds in a molecule increases its likelihood of being non-mutagenic (Class 1). Similarly, in the NCI1 dataset, *GOAt* discovers that the GNN considers a higher number of carbon rings as evidence of chemical

Table 1: Qualitative results of the top motifs of each class produced by PGExplainer, SubgraphX, RCExplainer and *GOAt*. The percentages indicate the coverage of the explanations.

| | BA-2Motifs | | Mutagenicity | | NCI1 | |
| | Class0 | Class1 | Class0 | Class1 | Class0 | Class1 |
|---|---|---|---|---|---|---|
| PGExplainer | 4.8% | 1.8% | 1.2% | 1.3% | 0.1% | 0.5% |
| SubgraphX | 0.4% | 12.8% | 0.2% | 0.2% | 0.2% | 0.1% |
| RCExplainer | 6.4% | 6.2% | 0.4% | 0.5% | 0.05% | 0.1% |
| *GOAt* | 3.8%  3.4% | 93.4%  4% | 3.5%  2.2% | 2.2%  1.2% | 3.5%  1.2% | 4.3%  4.0% |

compounds being active against non-small cell lung cancer. Other approaches, on the other hand, fail to provide clear and human-understandable explanations.

## 5 Related Work

Local-level Graph Neural Network (GNN) explanation approaches have been developed to shed light on the decision-making process of GNN models at the individual data instance level. Most of them, such as GNNExplainer [31], PGExplainer [17], PGM-Explainer [27], GraphLime [10], RG-Explainer [23], CF-GNNExplainer [16], RCExplainer [3], CF$^2$ [26], RelEx [34] and Gem [15], train a secondary model to identify crucial nodes, edges, or subgraphs that explain the behavior of pretrained GNNs for specific input samples. However, the quality of the explanations produced by these methods is highly dependent on hyperparameter choices. Moreover, these explainers' black-box nature raises doubts about their ability to provide comprehensive explanations for GNN models. Approaches like SA [4], Grad-CAM [20], GNN-LRP [22], and DEGREE [8], which rely on gradient back-propagation, encounter the saturation problem [24]. As a result, these methods may generate explanations that are less faithful. SubgraphX [33] combines perturbation-based techniques with pruning using Shapley values. While it can generate some high-quality subgraph explanations, its computational cost is significantly high due to the reliance on the MCTS (Monte Carlo Tree Search). Additionally, as demonstrated in our experiments in Section 4, existing approaches exhibit inconsistencies on similar data samples and poor discriminability. This reinforces the need for our proposed method *GOAt*, which outperforms state-of-the-art baselines on fidelity, discriminability and stability metrics. Our work also relates to global-level explainability approaches. GLGExplainer [2] leverages prototype learning and builds upon PGExplainer to obtain global explanations. GCFExplainer [11] generates global counterfactual explanations by employing random walks on an edit map of graphs, utilizing local explanations from RCExplainer and CF$^2$. Both GLGExplainer and GCFExplainer heavily rely on local explanations. Integrating local explainers that produce higher-quality local explanations, such as *GOAt*, has the potential to enhance the performance of these global-level explainers.

## 6 Conclusion

We propose *GOAt*, a local-level GNN explainer that overcomes the limitations of existing GNN explainers, in terms of insufficient discriminability, inconsistency on same-class data samples, and overfitting to noise. We analytically expand GNN outputs for each class into a sum of scalar products and attribute each scalar product to each input feature. Although *GOAt* shares similar limitations with some decomposition methods of requiring expert knowledge to design corresponding explaining processes for various GNNs, our extensive experiments on both synthetic and real datasets, along with qualitative analysis, demonstrate its superior explanation ability. Our method contributes to enhancing the transparency of decision-making in various fields where GNNs are widely applied.

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
