# OpenReview forum: "GOAt: Explaining Graph Neural Networks via Graph Output Attribution"
_NeurIPS.cc/2023/Conference — Submitted to NeurIPS 2023_

### Official Review · Reviewer_x34t · 2023-07-05

**Soundness:** 3 good
**Presentation:** 2 fair
**Contribution:** 3 good
**Rating:** 6
**Confidence:** 5

**Summary:**

This paper proposes a novel local-level post-hoc GNN explainer (GOAt) that takes the issue of discriminability and consistency into consideration.  The discriminability and consistency of graph explanation are significant and meaningful for the community’s understanding of the GNN decision-process. The GOAt explainer attributes the prediction of the trained GNN to the input graph features, i.e., the node feature matrix and the adjacency matrix, based on the Equal Contribution technique. The authors implement the specific GOAt with regard to GCN, GraphSAGE, and GIN. Sufficient experiments, including both qualitative and quantitative evaluation, verify that GOAt outperforms the SOTA GNN explainers in terms of both fidelity, discriminability, and stability.

**Strengths:**

S1: The issue of discriminability and consistency (i.e., stability) in GNN explanation methods is thought-provoking. However, it is neglected by previous methods. The GOAt explainer bridges the research gap.
S2: The mathematical derivation of the equal contribution technique is solid and ingenious, which well support the design of GOAt explainer.
S3: The experiments to verify the superiority of GOAt are sufficient and the results are discussed in detail.

**Weaknesses:**

W1: My main concern is about the effectiveness of the attribution strategy based on equal contribution. Though the experimental results have shown it outperformance, the intuitive reason why equal contribution can improve the discriminability and stability of GOAt explanation is still obscure for me. In other words, the motivation behind GOAt is not well-illustrated.
W2: The organization and writing of this manuscript need to be improved. Some contents are a little confusing. For example, in line 155, the authors claim that the number of occurrences is not necessarily equal to the number of unique variables. However, the example following does not explain this claim well.

**Questions:**

Q1: If GOAt take the node feature matrix X as constant finally (as stated in line 209), introducing this setup at beginning may be able to accelerate the GOAt inference. Moreover, why not take both node feature and topological structure as explanation, as done in GNNExplainer?
See WEAKNESSES for more detailed questions.

**Limitations:**

The authors have discussed some limitation about GOAt, including the requirement of expert knowledge, the failure in explaining deep models (e.g., Transformer). In my regard, as a post-hoc explainer, GOAt has to access the detailed architecture of the GNN model to be explained, which may serve as the main limitation in real-world application.

---

> ### Author Rebuttal · Authors · 2023-08-04
>
> * **(W1) The motivation behind GOAt in addressing the discriminability and inconsistency.**
>   * The existing approaches fail because most of them are trained or searched towards some human-defined metrics (such as explanation accuracy), which are not guaranteed to be faithful to GNNs. Other methods like the CAM methods require the gradients calculated with back-propagation, where the errors raised by the gradient saturation cannot be ignored. In contrast, GOAt forwardly computes the attribution without involving gradients or auxiliary models. This design choice allows our approach to be more faithful to the GNN itself, thus reducing the overfitting to noise issue or the gradient saturation issue. Furthermore, GOAt offers a choice to effectively handle discrete inputs, especially for the elements in the adjacency matrix, where the values can only be either 0 or 1, representing the absence or presence of an edge. Since the data with adjacency entries of real values between 0 and 1 do not exist in the dataset (i.e. out of distribution), we cannot assume values between 0 and 1 would carry cumulative account for the edges. In this context, the design of GOAt that treats 0 and 1 as two different states is more appropriate. This diverges from decomposition or gradient explainers, which consistently treat all inputs as continuous variables with cumulative implications.
>   * By assessing our newly introduced thought-provoking metrics discriminability and stability, as presented in Section 4.2, 4.3 and Appendix I,J,K, our approach demonstrates much stronger discriminatory capability between classes and consistency on the same-class data samples, compared with all the state-of-the-art baselines. Please also refer to the author's rebuttal titled "Response to all the reviewers" for clarifications on why existing approaches fail in discriminability and consistency, and how our approach addresses these issues.
>
> * **(W2) Typos, misunderstandings in the manuscript.**
>   * We will modify our manuscript by adding the clarifications in the author's rebuttal titled "Response to all the reviewers" and replace the complicated examples like the one in line 155 to simple ones to help the reader better comprehend our paper. In particular, the example in Line 155 means: if a product term is $z=x^2y$, then the number of variables $|V(z)|=2$, where ${x,y}$ are the two variables. And the total number of occurrences of all the variables is $O(x,z)+O(y,z)=2+1=3$. These two values are not equal.
>
> * **(Q1) If GOAt take the node feature matrix X as constant finally (as stated in line 209), introducing this setup at beginning may be able to accelerate the GOAt inference. Moreover, why not take both node feature and topological structure as explanation, as done in GNNExplainer?**
>   * We will incorporate your suggestion into our manuscript by clarifying that only entries in matrix $A$ are treated as variables at the beginning of Section 3. Thank you very much.
>   * The reason we didn't include node features and also topological structure as explanations, is explained in Line 69. Unlike GNNExplainer's explanations that encompass both node features and topological structure, which can be challenging for humans to comprehend, our goal is to provide understandable explanations through subgraphs, a strategy employed in [33,23,8].
>   * Edges offer more precise insights than nodes [7], contributing to our aim of human-friendly explanations.
>     * For example, there could be potential mismatch between nodes with "important" features and nodes with "important" direct edges, because some nodes gain significance from their topological context rather than their features. Then it becomes hard to decide when we should encounter a node to the explanation.
>     * Furthermore, picking nodes with important node features can result in isolated nodes in the explanation, which  is not desirable.
>     * To ensure human-understandable subgraph explanations and assess the fidelity, discriminability, stability, we need to extract subgraphs with consistent criteria. As highlighted by [7], edges are superior to nodes in this context, which is why we use only the edges to extract the explanation motifs.
>
> We hope these clarifications address your concerns. Your feedback is greatly appreciated.

---

> > ### Comment · Reviewer_x34t · 2023-08-15
> >
> > Thanks for the authors' response to my questions. My most concerns are well addressed. I will keep my score as weak accept.

---

> > > ### Author Response · Authors · 2023-08-17
> > >
> > > Many thanks for your positive remarks. Thank you for the feedback to help enhance the quality of our research.

---

### Official Review · Reviewer_b4jf · 2023-07-05

**Soundness:** 2 fair
**Presentation:** 3 good
**Contribution:** 2 fair
**Rating:** 5
**Confidence:** 3

**Summary:**

In this paper, the local explanation algorithm for Graph Neural Networks (GNN), Graph Output Attribution (GOAt), is proposed. The key idea of the algorithm is decomposing the GNN's forward propagation as sum of product form. After decomposing, authors compute the contributions of nodes and edges by proposed measure. Next, (sum of the contribution including the interest node or edge) - (contribution from global bias) becomes the contribution (explanation) of the node or edge. The experiments show (1) the explanation can capture the importance of edges (fidelity, discriminability) (2) the explanation is consistence (Stability).

**Strengths:**

1. The algorithm is general for graph neural networks.
2. The examples help the reader understand the explanations.
3. The experiments show the performance of proposed model with regard to baselines for various criteria.

**Weaknesses:**

1. The paper used zero-vector as base manifold. It makes that the contributions for all variables are always equal.
2. The contribution does not consider the value of variables.
3. It would be better to explain the difference between the baseline algorithms and proposed algorithm.

**Questions:**

1. What is motivation of setting base manifold as zero-vector? It seems to make equal contribution, but I think the other resonable explanation is necessary.
2. When we consider the 3 variables (x1',x2',x3'), the manifolds at (x1,x2,x3) are {(x1,x2',x3'), (x1',x2,x3'), (x1',x2',x3)}? If so, why {(x1,x2,x3'),(x1,x2',x3),...} are not considered like shapley value?
3. In lemma 4, why "when the number of nodes N is large" is necessary?
4. In line 144, the |V(x^2y)|={x,y}, this makes x and y have same contribution for x^2y. It is not reasonable.
5. In line 154, it is hard to understand O(v,z) (the number of occurrences of ν among the variables of z).
6. In line 217, I think it is unfair to compare the proposed edge explanation and the edge explanation from node explanations of baselines by [3]. Because the proposed algorithm can compute the node explanation like baselines, same process [3] should be used for edge explanation from proposed node explanation.
7. In figure 3 (h), the plot seems like there are less points than others.
8. In figure 3, what kinds of algorithms is used for 2D plotting? How the position of green scatter points is consistent for various embeddings? I think the different 2D projections should be used for each embeddings.
9. In line 282, why the figure 3 (h) explains the data samples use only 3 explanation? If 3 explanations used, I think 2^3 points (existence of each explanation) should exist in figure 3.
10. In table 1, there is no motifs for "This indicates that the presence of more carbon rings in a molecule increases
 its likelihood of being mutagenic (Class 0)" (line 293) and "GOAt discovers that the GNN considers a higher number of carbon rings as evidence of chemical compounds being active against non-small cell lung cancer" (line 296)

(minor comment) there is no definition for the j-th feature of i-th node (line 126)

**Limitations:**

The paper used zero-vector as base manifold. It makes that the contributions for all variables are always equal. Also, the proposed contribution consider absence or presence of variable only without the value of variable.

---

> ### Author Rebuttal · Authors · 2023-08-05
>
> * (**W1, Q1**) **Why is zero-vector used in the _Equal Contribution_ algorithm?**
>     * The elements in the adjacency matrix can be {0,1}, indicating the absence or presence of edges between pairs of nodes. Any values between {0,1} are actually not relevant to the problem, as they do not carry any accumulative meaning. Therefore, it is crucial to select zero vector as the base instead of any other values between {0,1}.
>     * "Equal contribution" plays an essential role in the scalar product expansions of the output, it is not a weakness but a fundamental aspect of our approach. The design of the _Equal contribution_ algorithm enables it to handle discrete inputs more effectively, and mitigate the saturation problem that almost all other attribution approaches suffer from.
>     * For more detailed information on how our algorithm works and why the equal contribution and base are used, I kindly refer you to Section 3 of our paper and the second and third point in our rebuttal "Response to all the reviewers." It provides a comprehensive explanation of our approach and its rationale.
>
> * (**W1, Q4**) **Does GOAt makes that the contributions for all variables are always equal?**
>     * No. Each edge participates in only some of the scalar products. Various scalar products have various values. Therefore, the contributions for all variables cannot be always equal. Please refer to Section 3 of our paper and our rebuttal "Response to all the reviewers" for more details.
>
> * (**W2**) **"The contribution does not consider the value of variables."**
>     * The value of variables are considered in our algorithm. The larger a variable is, the larger the product that it participates would be. Each edge participates in many the scalar products. If its value is larger, then the resulting scalar products will be larger. Thus, its contribution to these scalar products will be larger too.
>
> * (**W3**) **Difference between the baseline algorithms and proposed algorithm?**
>    * Our approach is completely different from the baselines. As we summarised in Section 5, most of the existing models train auxiliary models to learn masks for the explanations, where the black-box nature of them raises doubts about their faithfulness. Other methods either rely on inefficient search or back-propagation gradients, where the former can be computationally costly and the latter suffers from the saturation problem. On the other hand, GOAt avoids the training of auxiliary models and forwardly compute the attribution hence can be more efficient, mitigate the saturation problem and provide more comprehensive explanations. Furthermore, the baseline algorithms suffer from numerous drawbacks, including poor discriminability, inconsistency, and overfitting to noise, while GOAt achieves better discriminability and consistency through extensive experiments in our paper. Please also refer to our rebuttal "Response to all the reviewers" for details.
>
> * (**Q2**) **Why not use Shapley value?**
>    * Our approach is entirely distinct from Shapley value, and applying Shapley value to the numerous "scalar products" in the expansion form of the output is not computationally feasible. For a comprehensive understanding of our algorithm, please refer to our rebuttal titled "Response to all the reviewers" which elaborates on how our approach operates in detail.
>
> * (**Q3**) **In lemma 4, why "when the number of nodes N is large" is necessary?**
>    * Just like many theoretical proofs, we simply use this assumption to provide theoretical analysis and mathematically prove the feasibility and reasonableness of the algorithm. The empirical results presented in Section 4 further demonstrate the effectiveness of our algorithm in practical situations. Namely, in practice, when the number of nodes is not that large, we can still effectively explain the GNN with the Equal contribution assumption, and achieve superior explaining performance compared with the state-of-the-art baselines.
>
> * (**Q5**) **What is O(v,z)?**
>    * As clarified in Line 154, $O(v,z)$ refers to the number of occurrences of $v$ in $z$. For example, if $z=x^2y$, then $O(x,z)=2$, $O(y,z)=1$.
>
> * (**Q6**) **Why not use GOAt to generate subgraph explanations with nodes, when comparing with node-selection approaches?**
>    * In the experiments, we compare the best setup of each algorithm. As pointed out in [7], edge-selection provides more fine-grained information compared to node-selection. Therefore, all algorithms that allow edge-selection should use the edge-selection setup, which aligns with [3].
>
> * (**Q7, Q8, Q9**) **Why are there 3 points on the scatter plot instead of 8? What is the algorithm for scatter plots?**
>    * Figure 3 shows significant overlapping of points, resulting in the observation of only three points.
>    * In order to visualize the hidden representation before classification in a lower dimension, we employ PCA.
>    * Notably, identical explanations yield the same embeddings. Consequently, the identical explanation motifs are associated with identical points on the scatter plot.
>
> 9. (**Q10**) **Where is the carbon ring in Table 1?**
>    * In Table 1, the presence of multiple C-C-C motifs indicates the occurrence of carbon rings when only a small portion of edges is allowed for the explanation, aligning with the findings in [2].
>
> We hope these clarifications help your comprehend our paper. We kindly request that you consider revising the score based on the additional information provided. Your feedback is greatly appreciated.

---

> > ### Comment · Reviewer_b4jf · 2023-08-18
> >
> > Thanks to the authors for their responses.  Many minor concerns and questions are responded by authors.  I can not fully agree with that opinion, “zero-vector as base manifold”, but the point of that opinion is understandable. I think it is similar problem like “what is the base image in image data”. Someone can say white image like this authors, and the others can say gray or black or expectation of pixels and so on. I think it would be better if the authors could support their opinions clearly. Since the responses addressed most of my concerns, I change my score to 5: Borderline accept.

---

> > > ### Author Response · Authors · 2023-08-19
> > > **Thank you very much for the feedback.**
> > >
> > > Yes. It is similar to "the base image in image data". As the reviewer noted, in computer vision domain, someone can say white image as the base, and others can say grey or black or expectation of pixels as the base. They want to explain the networks with respect to a "base" which carries no meaningful information of the target class, highlighting the features or aspects of the image that the neural network has learnt to focus on. Similarly, an edge in a graph can be either present (1 in the adjacency matrix) or absent (0 in the adjacency matrix) as we defined in the problem formulation. For edges in a graph, we can naturally set the base to zero-vectors (to avoid carrying specific topological features to a target class), based on which GOAt computes edge contributions to the output. The experimental results on fidelity, discriminability, and stability have demonstrated the effectiveness of attributing the graph output to edges by GOAt. We agree with the reviewer that explaining in this way could be clearer, and we will follow the reviewer's suggestion by incorporating these notes into the paper.
> > >
> > > Many thanks for the feedback and opportunity to help enhance the quality of our paper. Please let us know if you have any further concerns.

---

### Official Review · Reviewer_LXep · 2023-07-06

**Soundness:** 3 good
**Presentation:** 2 fair
**Contribution:** 3 good
**Rating:** 6
**Confidence:** 4

**Summary:**

It proposes a local GNN explanation method called GOAt by analysing the output attribution of graphs. Specifically, it proposes a novel explanation framework by the notion of equation contribution, which is transparent compared to the existing black-box type explanation models. Experiments on real and simulation data are tested on several metrics, where the results are convincing.

**Strengths:**

1. The point of equation contribution is quite novel and interesting. Especially its transparency compared to current baselines.
2. The experiments are extensive and convincing.

**Weaknesses:**

1. In line 28, it seems that "their effectiveness relies heavily on the quality of local-level explanations" is the main point of abandoning the global style in this paper, could you explain more about this in the rebuttal phase or say more about it in section 2 - see second con below.
2. In section 2, the differences between global and local explanatory methods should be introduced in a proper way to show the superiority of global explanations. Also, please consider including the matrices in lines 66-69 as another subsection to introduce the matrices together.

**Questions:**

1. Major:
    1. In lines 103-107, the reasonableness of the independence between the variables $X$ and $A$ in the contribution for $f(A,X)$ should be further improved. As I understand it, the equal contribution property of $f_{i,j}(A,X)=\sum_k\sum_l A_{i,k} X_{k,l} W_{l,j}$ is not identical to $A_{i,k} X_{k,l} W_{l,j}$, because the distribution of $A$ can never be independent of $X$. In line 103 the authors mention that "since $A_{i,k}$ is usually independent of $X_{k,l}$ under proper data cleaning" - this is true, while it is not equal to $\sum_k\sum_l A_{i,k} X_{k,l} W_{l,j}$.
    2. Could you please explain more about the reduction of equation (7)?
    3. For the proof of Lemma 4 and Theorem 5, what's the use of the condition requiring a large number of nodes? It is highly recommended that the authors make this condition more practical or reanable in real-world cases.
2. Minor:
    1. In lemma 2, better include the condition of $x_k \neq 0$;
    2. Is the number of variables $|V(z)|$ of equation (8) equal to the number of parameters?
    3. I'm also wondering if there is a gap between node-based and edge-based explanation methods, since in the experiments the node-based approaches are aligned with the edge-based ones by using their induced subgraphs. Can the authors explain more about the correct explanation is symmetric in both types of methods, or provide some basic intuition of this?
    4. In lines 173-177 the authors mention the property of completeness. A question here is whether in the experiments the top-k edges are still selected to compute the evaluation matrices?
    5. Following the idea of matrix-based understanding of GNNs, there is a recent paper you might consider citing: https://arxiv.org/abs/2305.06142 Feature expansion for graph neural networks. J Sun, L Zhang, G Chen, P Xu, K Zhang, and Y Yang. ICML'23. You can use it to extend your framework to more GNN approaches.

**Limitations:**

The authors presented the limitations in the Appendix about wider range of model architectures.

---

> ### Author Rebuttal · Authors · 2023-08-04
>
> * **(W1, W2) Should explain the differences between global-level explanations and local-level explanations and show the superiority of global-level explainability in the related work and problem formulation.**
>
>   Global-level GNN explainability is a parallel sub-domain to the local-level explainability that our paper primarily focuses on.
>   * The challenges faced by global-level explainers differ from those encountered at the local level. While the local-level explanations aim to accurately explain each data instance, global-level explanations encompass the overall behavior of a complex model, across the entire dataset or model space.
>   * Global-level explainability offers more human-friendly explanations, such as boolean rules or counterfactual rules. They provide insights into how the model functions on a broader scale, often revealing general patterns, trends, or relationships within the data. These explanations can offer a high-level understanding of the model's decision-making process and its interactions with various features or components.
>   * Some existing global-level explainers build on local-level explainers and construct models to extract global rules. In this context, enhancing the quality of local explanations, for instance, by utilizing our proposed GOAt, has the potential to improve the performance of global-level explainers.
>
> * **(Q1.1) The example in lines 103-107 is not mathmetically accurate.**
>   * I see your point. However, the equation and discussion In Line 103-107 are not intended to be theoretical but rather serve as a toy example to provide a concise overview of our approach and aid readers' comprehension of the subsequent theorems. We will make adjustments to the terminology, using "ideal data cleaning" instead of "proper data cleaning" to convey the independence of $A_{i,k}$ and $X_{k,l}$. Would you consider this modification more appropriate?
>
> * **(Q1.2) Could you please explain more about the reduction of equation (7)?**
>   * Sure. In the explanation mode, the parameters of the GNN, such as weights $W$, biases $B$, and coefficient $\lambda$, remain fixed as they were pretrained. As a result, they can be treated as constants in the scalar product terms. Hence, we simplify and denote them as $\mathbb{C}$, which represents the constant terms in the scalar product expressions.
>
> * **(Q1.3) For the proof of Lemma 4 and Theorem 5, what's the use of the condition requiring a large number of nodes? Why not make mathematical proofs in more practical or real-world cases?**
>   * Just like many theoretical proofs, we provide theoretical analysis results under the large number assumption and mathematically establish the feasibility and reasonableness of the proposed algorithm. However, a large number of nodes is not required in practical application of GOAt. The empirical results in Section 4 validate the effectiveness of our algorithm in practical scenarios and show that when the number of nodes is not very large, we can still effectively explain the GNN with the Equal Contribution assumption and achieve superior explaining performance compared to various state-of-the-art baselines.
>
> * **(Q2.2) What is $|V(z)|$ in equation (8)?**
>   * It is the number of variables, where variables can be $X_{ij}$, $A_{cd}$, $P_{ef}$.
>
> * **(Q2.3) Is there a gap between node-based and edge-based explanation methods? Are the node-based approaches aligned with the edge-based ones?**
>   * As shown in Figure 1, there is a noticeable performance gap between the node-based approach (SubgraphX) and the other edge-based approaches. This gap can be attributed to the difficulty node-based methods face in accurately determining the critical edges around the "corner" critical nodes of the induced subgraph. This limitation results in less fine-grained explanation capability compared to edge-based methods, as also discussed in [7]. Due to this fundamental difference in their explanatory nature, we believe that node-based approaches cannot fully align with the edge-based ones.
>
> * **(Q2.4) Given the completeness, are the top-k edges are still selected in the experiments?**
>   * Passing the sanity check of completeness confirms that our approach provides a comprehensive account of the attribution, while most of the existing learning-based or search-based approaches do not. In the experiments, we select the top edges at different sparsity levels to align with the baselines [31,17,27,33,16,3,23,8]. Compared with the baselines that select top edges without satisfy the completeness, GOAt is more statistically convincing.
>
> * **(Q2.5) Feature expansion paper in improving GNN's performance.**
>   * Thank you for sharing this paper. We find the discoveries regarding the linear correlation in the feature space due to repeated aggregations to be intriguing and thought-provoking. The empirical results showcasing its effectiveness in enhancing GNN performance are commendable. It's gratifying to observe that exploring the matrix behavior of GNNs not only contributes insights into explaining GNNs, as demonstrated in our GOAt approach, but also aids in the advancement of GNN architecture. We will cite this paper.
>
> Thank you for pointing out some minor typos. We will revise the paper accordingly.
>
> We hope these clarifications address your concerns. We kindly request that you consider revising the score based on the additional information provided. Your feedback is greatly appreciated.

---

> > ### Comment · Reviewer_LXep · 2023-08-14
> > **Thanks for the rebuttal**
> >
> > Thanks to the authors for their detailed responses. The minor concerns are mostly well addressed and the theoretical requirements of a large number of nodes are also noted. However, two questions/suggestions remain.
> >
> > 1. (W1&W2): This paper actually proposes another novel local level explicator equipped with several ideal characters, which global types also require. Thank you for your explanation on this - it is fatal, and I strongly recommend the authors to rearrange the introduction and literature review parts to include this connection to make the motivation clearer. For example, in section 2, talk about global type and make comparisons.
> >
> > 2. (Q 1.1): this is not just a question of terminology, but of the mathematical definition that the whole paper will follow. If the independent feature is not important, you can just omit it; otherwise, make it clear, e.g. that this is a theoretically friendly way and will not affect the empirical results, or give some solid hint that this assumption works.
> >
> > Thanks again for the rebuttal, and I am happy to raise the score to 6. Please the authors take the issue of motivation seriously.

---

> > > ### Author Response · Authors · 2023-08-17
> > >
> > > 1. **(W1&W2)**: We will incorporate your suggestions and add the comparison of global-level and local-level explainers to both Section 1 (Introduction) and Section 5 (literature review). Additionally, we will distinguish global-level and local-level explanations in Section 2, problem formulation, by examples, and note the connection between the two.
> > > 2. **(Q 1.1)**: Thank you very much for the suggestion. We will make a note in Line 103, noting that "To be theoretically friendly, we assume $A_{i,k}$ is independent of $X_{k,l}$, this assumption will not affect the empirical results."
> > >
> > > Many thanks for your positive remarks. Thank you for the feedback and please feel free to let us know if you have any further concerns.

---

### Official Review · Reviewer_v7M6 · 2023-07-24

**Soundness:** 3 good
**Presentation:** 2 fair
**Contribution:** 3 good
**Rating:** 5
**Confidence:** 4

**Summary:**

Unlike existing methods that rely on training auxiliary models, the paper introduces a computationally efficient instance-level GNN explanation technique (called GOAt), which enables attribution of GNN output to input features by direct algebraically computation. More specifically, a GNN is represented in an expansion form as a sum of scalar product term, involving input graph features, model parameters, and activation levels. Based on the assumption that all scalar variables in a scalar product term contribute equally to the term, each product term can be attributed to its corresponding factors and thus to input features, which gives the importance of each node or edge feature in the input graph to GNN outputs. Besides that, the paper also introduce two new metrics called discriminability and stability, which assess the ability of explanations to distinguish between classes and the ability to generate consistent explanations across similar data instances, respectively. Through extensive experiments on synthetic and real data, the paper show that GOAt has consistently outperformed various state-of-the-art GNN explainers in terms of fidelity, discriminability, and stability.

**Strengths:**

1.	The assumption that all scalar variables in a scalar product term contribute equally to the term is acceptable to me. And the derived lemmas and theorems are well proved and practically applicable.
2.	The proposed explanation method is computationally efficient, and does not rely on back-propagation with gradients, hyper-parameters or training complex black-box models, which is novel to me.
3.	The work contributes a new attribution method to the field of explaining GNNs (might extend to other types of neural networks, e.g., CNN) .


**Weaknesses:**

1.	The specific calculation expression of the attribution process needs to be carefully derived with expert knowledge for different neural network architectures and becomes very complex as the network goes deeper, which greatly limits its widespread use, though it shows superior explanation ability.
2.	The main argument that existing methods struggle with issues such as overfitting to noise, insufficient discriminability, and inconsistent explanations across data samples of the same class is not well elaborated. Though the experiments seem to have verified this argument, it will be more motivated if some detailed illustrations (e.g., by presenting a toy example) could be provided in the introduction part.


**Questions:**

1.	The assumption that the number of nodes of the input graph is large in the key lemma(Lemma 4) and the theorem(Theorem 5) is not well established in the datasets used in experiments.

**Limitations:**

a)	Build an algorithm to automatically derive the attribution expressions based on the network architecture.
b)	Although the experiments seem to corroborate the main argument, bolstering the introduction with some detailed illustrations (for instance, by incorporating a simple, illustrative example) could make the paper more compelling.

---

> ### Author Rebuttal · Authors · 2023-08-04
>
> * **(W1) The explainer can become very complex as the network goes deeper.**
>   * Our paper only focuses on the Graph Neural Network explainability instead of the general neural network explainability. As outlined in the "Limitations and Broader Impacts" section, our proposed approach can effectively explain GNNs due to their relatively shallow structure. While the fundamental idea of "Equal Contribution in the Scalar Product" could potentially be extended to elucidate deeper neural networks other than GNNs, it would require the grouping or pruning of scalar products that have minimal impact on the outputs. It would be a future work for the community.
>
> * **(W2) Should introduce more about why existing methods struggle with issues such as overfitting to noise, insufficient discriminability, and inconsistent explanations across data samples of the same class in the introduction part.**
>   * For sure. We will include more intuitions on why the existing approaches fail to address these issues.
>   * Specifically, the existing approaches fail because most of them are trained or searched towards some human-defined metrics (such as ground-truth accuracy or fidelity), which are not guaranteed to be faithful to GNNs. Other methods like the CAM methods require the gradients calculated with back-propagation, where the errors raised by the gradient saturation cannot be ignored. In contrast, GOAt forwardly computes the attribution without involving gradients or auxiliary models. This design choice allows our approach to be more faithful to the GNN itself, thus reducing the overfitting to noise issue or the gradient saturation issue. By assessing our newly introduced thought-provoking metrics _discriminability_ and _stability_, as presented in Section 4.2, 4.3 and Appendix I,J,K, our approach demonstrates much stronger discriminatory capability between classes and consistency on the same-class data samples, compared with all the state-of-the-art baselines.
>   * For clarifications on why existing approaches struggle with discriminability and consistency, as well as how our approach resolves these issues, we kindly direct you to the author's rebuttal under the title "Response to all the reviewers."
>
> * **(Q1) The assumption that the number of nodes of the input graph is large in the key lemma(Lemma 4) and the theorem(Theorem 5) is not well established in the datasets used in experiments.**
>   * Just like many theoretical proofs, we provide theoretical analysis results under the large number assumption and mathematically establish the feasibility and reasonableness of the proposed algorithm. However, a large number of nodes is not required in practical application of GOAt. The empirical results in Section 4 validate the effectiveness of our algorithm in practical scenarios and show that when the number of nodes is not very large, we can still effectively explain the GNN with the Equal Contribution assumption and achieve superior explaining performance compared to various state-of-the-art baselines.
>
> We hope these clarifications address your concerns. We kindly request that you consider revising the score based on the additional information provided. Your feedback is greatly appreciated.

---

> > ### Comment · Reviewer_v7M6 · 2023-08-18
> > **Response to author.**
> >
> > Thank you for your response. These have addressed the majority of my questions. I appreciate the effort and insights the authors put into the paper. I will maintain my original score.

---

### Official Review · Reviewer_33zp · 2023-07-27

**Soundness:** 2 fair
**Presentation:** 2 fair
**Contribution:** 2 fair
**Rating:** 5
**Confidence:** 3

**Summary:**

This work studies the problem of GNN local explanation.  It points out that existing GNN explainers are suffering from a few limitations including insufficient discriminability, inconsistency on same-class data samples, and overfitting to noise, and the aim to address these limitations by proposing the GOAt method. Given a pre-trained GNN model, GOAt defines a contribution score on each node feature and each edge feature (i.e. values on each edge) by expanding the GNN as a sum of scalar products involving these features, and thus would be able to find the edges that contribute more to the GNN's output. The authors evaluate GOAt with the frequently used fidelity metric, as well as the newly proposed discriminability and stability metric in this paper.

**Strengths:**

1. Results are impressive.
2. Code is provided in the supplementary.
3. The newly introduced discriminability and stability metric for explanation evaluation makes sense to me.

**Weaknesses:**

1. The writing for the method section is not easy to follow.
2. The motivation of why the contribution should be designed in this way is not clearly explained. Why the current design can address the insufficient discriminability, inconsistency on same-class data samples, and overfitting to noise issues also remains unclear to me.

Some minor weaknesses:
1. There seems to be one typo in the appendix proof lemma 2: line 11 should be X_j not E_j?
2. The term ``feature`` is used to describe all X_{ij}s and A_{ij}s, which looks a little bit confusing for me. I think usually people use the term ``feature`` to describe the attributes on nodes and edges, but here A_{ij} is more like a weight scaler on edge instead of the edge attributes? I feel it would be clearer to put a note to clarify that the term ``feature`` is used to describe all node attributes and edge weights.
3. It would be clearer if the authors can explain what does each lemma and theorem imply.

**Questions:**

Q1. I am still very confused about, why the contribution should be defined in this way? In other words, what does this formula imply and why it can act as an evaluation of contribution?

Q2. Why GOAt can achieve better discriminability?

Q3. What is the scope of GNN designs that can be explained by GOAt? (For example, is every GNN model that can be formulated in the form of formula (1) covered in the scope?)

**Limitations:**

Please refer to ``weaknesses``.

---

> ### Author Rebuttal · Authors · 2023-08-04
>
> * (**W2 & Q1**) **How is the contribution of an edge determined?**
>    * _Equal Contribution_:
>       * Consider a product term $z = 10A_{11}A_{12}A_{23}$, where the variables $A_{11} = A_{12} = A_{23} = 1$ indicate that all three edges exist, resulting in $z = 10$. If any of the edges is missing (at least one of $A_{11}, A_{12}, A_{23}$ is $0$), then $z = 0$.
>       * This implies that the presence of all edges is equally essential for the resulting value of $z = 10$. Therefore, each of the three edges contributes $\frac{1}{3}$ to the output, resulting in an attribution of $\frac{10}{3}$.
>
>    * _Expand the output representation_:
>        * The output matrix of a GNN can be described as the outcome of a linear transformation involving the input matrices and the GNN parameters. Since the GNNs are pretrained, the parameters $W,B$ are fixed, hence can be treated as constants. As a result, each element within the output matrix can be represented as the sum of scalar products that involve entries from the inputs. Consequently, we can determine the attribution of an edge, by summing its contribution across all the scalar products in which it participates.
>       * For example, if the expansion form of an element in the output matrix is $y_{13}=10A_{11}A_{12}A_{23} + 8A_{11}A_{13}A_{33} + 11A_{12}A_{21}A_{13}$, then the attribution of $A_{11}$ to $y_{13}$ is $10/3+8/3=6$ since $A_{11}$ does not participate in the term $11A_{12}A_{21}A_{13}$.
>
> * (**W2 & Q2**) **Why does our algorithm achieve better discriminability and consistency compared with existing approaches?**
>     * Existing approaches struggle with discriminability and stability as they train auxiliary models to optimize predefined metrics, leading to overfitting issues. For example, optimizing for fidelity metrics can result in overfitting to noise, as removing noise may artificially improve the fidelity score. Noise in the graph refers to the edges that may improve the fidelity metric, but not relevant to the original GNN's prediction. The noise lacks discriminatory power between classes, also leading to explanations that are inconsistent on same-class data samples. In contrast, GOAt directly computes the attribution of each edge to the GNN prediction, without the need for auxiliary models. This design choice allows our approach to be more faithful to the GNN itself, thus reducing susceptibility to overfitting to noise. By assessing our newly introduced thought-provoking metrics _discriminability_ and _stability_,  as presented in Section 4.2, 4.3 and Appendix I,J,K, our approach demonstrates much stronger discriminatory capability between classes and consistency on the same-class data samples, compared with all the state-of-the-art baselines.
>
> * (**Q3**) **Are all the GNN models that can be formulated in the form of Equation (1) covered in the scope?**
>     * Yes, GNN models that can be formulated in the form of Equation (1) are covered in the scope. As we explained in Section 3.2, the hidden state can be represented by Equation (4). Then we can go ahead and follow the formulas in Section 3.2 to obtain the attribution of each edge. In particular, we have demonstrate in details that how the typical GNNs including GCN, GIN and GraphSAGE are represented in the expansion form. The experimental results presented in Section 4 and Appendix showcase the superiority of GOAt across a variety of datasets including both synthetic datasets as well as real-world datasets.
>
> Thanks for pointing out the typo and the misunderstanding raised from “feature”. We will modify the manuscript accordingly.
>
> We hope these clarifications address your concerns. For detailed information, please refer to Line 21-29 Section 3 and the author's rebuttal titled “Response to all the reviewers”. We kindly request that you consider revising the score based on the additional information provided. Your feedback is greatly appreciated.

---

> > ### Comment · Reviewer_33zp · 2023-08-18
> >
> > Thanks to the authors for the response to my concerns. My major concerns are well addressed. I'm willing to increase my score to 5.

---

### Author Rebuttal · Authors · 2023-08-04

### **Response to all the reviewers**
Thanks to all reviewers for the constructive feedback. First, we would like to clarify our contribution as compared to the existing literature of GNN explanation:
1. Existing GNN explaining methods suffer from issues of **insufficient discriminability** between classes, and **inconsistency** on same-class data samples as they rely on auxiliary models to optimize human-defined metrics:

    First, optimizing the explainers for fidelity metrics or max mutual information can result in overfitting to noise. For example, removal of noise (instead of key component) may also artificially influence GNN confidence and thus affect fidelity scores. However, the noise in a graph lacks the ability to distinguish between classes, leading to spurious explanations that may vary according to samples even if they come from the same class, thus the inconsistent explanations on same-class data samples. Second, optimizing the explainers for the explanation accuracy is problematic. GNNs are trained to predict data labels, hence might not consistently be able to capture the "ground-truth explanations" that human comprehends. Consequently, these explanations might fail to differentiate between samples of different classes as classified by the actual GNN.

    Therefore, we introduce the discriminability and stability metrics to bridge the missing gap in the GNN explainability domain that has been overlooked by prior literature. As presented in Sections 4.2, 4.3 and Appendix I,J,K, our approach demonstrates much stronger discriminatory capability between classes and consistency on the same-class data samples, compared with a range of state-of-the-art baseline methods.

2. **The strengths of the proposed algorithm GOAt lie in the following**:
    * _Faithful to GNN_: It avoids the training of auxiliary models or optimization to some human-defined metrics (like ground-truth accuracy or fidelity) and directly computes the attribution of each edge to the GNN prediction, which allows GOAt to be more faithful to the GNN itself, mitigating the overfitting to noise issue hence have better discriminative capability and stability across the same-class samples.
    * _Mitigating saturation problem_: By forwardly computing the attribution without involving gradients, our approach does not suffer from the saturation problem that almost all other attribution approaches suffer from, including Sensitivity Analysis, Grad-CAM, DeepLift, Integrated Gradients, etc.
    * _Ability to handle discrete inputs_: The nature of GOAt algorithm enables it to handle discrete inputs more effectively, compared with methods like Sensitivity analysis and Integrated Gradients that always consider the inputs as continuous variables. For example, the elements in the adjacency matrix can be {0,1}, indicating the absence or presence of edges between pairs of nodes. Any values between {0,1} are actually not relevant to the problem, as they do not carry any accumulative meaning.
    * _Statistically convincing_: GOAt passes the sanity check that the attribution scores for input features add up to the difference in the GNN’s output with and without those features, whereas most of the search-based approaches or learning-based approaches cannot.


3. **High-level Algorithm Overview**:
    * _The Definition of Equal Contribution_:
        * Consider a product term $z = 10A_{11}A_{12}A_{23}$, where the variables $A_{11} = A_{12} = A_{23} = 1$ indicate that all three edges exist, resulting in $z = 10$. If any of the edges is missing (at least one of $A_{11}, A_{12}, A_{23}$ is $0$), then $z = 0$.
        * This implies that the presence of all edges is equally essential for the resulting value of $z = 10$. Therefore, each of the three edges contributes $1/3$ to the output, resulting in an attribution of $10/3$.
    * _Expansion form of the output representation_:
        * The output matrix of a GNN can be described as the outcome of a linear transformation involving the input adjacency/feature matrices ($A,X$) and the GNN parameters ($W,B$). As a result, each element in the output matrix can be represented as a sum of scalar products that involve entries from both the input matrices and the GNN parameters.
        * Each edge appears in only some of the scalar products. Consequently, we can determine the attribution of an edge, such as $A_{11}$, by summing its contribution across all the scalar products in which it participates.

We hope that our clarifications effectively address your concerns, and showcase the value of our algorithm and the newly introduced metrics.

---

### Comment · Area_Chair_m27Z · 2023-08-17
**AC discussions**

There are mixed reviews for this paper.
Reviewers: could you read author rebuttals and let us know if your concerns have been addressed?

AC

---

### Decision · Program_Chairs · 2023-09-21

**Decision:**

Reject

**Comment:**

This is a borderline paper according to reviewer recommendations and comments. The SAC and AC checked and read this paper during SAC-AC discussions. Two major issues arise during the discussions: 1. There are only a small number of datasets used in the experiments while prior work used much larger numbers of datasets. Given it is not very time-consuming to include more datasets, we felt the experiments are weak. 2. The authors proposed some new evaluation metrics, which may lack thorough justifications. In addition, the authors at least should use common metrics used in prior work. For example, they only used fidelity+, but not fidelity-, which is not common. Thus we feel the paper needs more work.